# Cone-Beam Computed-Tomography-Derived Augmented Fluoroscopy-Guided Biopsy for Peripheral Pulmonary Nodules in a Hybrid Operating Room: A Case Series

**DOI:** 10.3390/diagnostics13061055

**Published:** 2023-03-10

**Authors:** Lun-Che Chen, Shun-Mao Yang, Shwetambara Malwade, Hao-Chun Chang, Ling-Kai Chang, Wen-Yuan Chung, Jen-Chung Ko, Chong-Jen Yu

**Affiliations:** 1Interventional Pulmonology Center, National Taiwan University Hospital Hsin-Chu Branch, Hsinchu County 302, Taiwan; 2Department of Internal Medicine, National Taiwan University Hospital, Hsin-Chu Branch, Hsinchu County 302, Taiwan; 3Department of Surgery, National Taiwan University Hospital, Hsin-Chu Branch, Hsinchu County 302, Taiwan; 4Department of Advanced Therapies, Siemens Healthcare Limited, Taipei City 11503, Taiwan

**Keywords:** lung cancer, diagnostic yield, endotracheal tube, biopsy, nodules, percutaneous, lesions, bronchoscopy

## Abstract

Lung cancer is the most lethal cancer type in Taiwan and worldwide. Early detection and treatment advancements have improved survival. However, small peripheral pulmonary nodules (PPN) biopsy is often challenging, relying solely on bronchoscopy with radial endobronchial ultrasound (EBUS). Augmented fluoroscopy overlays the intra-procedural cone-beam computed tomography (CBCT) images with fluoroscopy enabling real-time three-dimensional localization during bronchoscopic transbronchial biopsy. The hybrid operating room (HOR), equipped with various types of C-arm CBCT, is a perfect suite for PPN diagnosis and other interventional pulmonology. This study shares the single institute experience of EBUS transbronchial biopsy of PPN with the aid of augmented fluoroscopic bronchoscopy (AFB) and CBCT in an HOR. We retrospectively enrolled patients who underwent robotic CBCT, augmented fluoroscopy-guided, radial endobronchial ultrasound-confirmed transbronchial biopsy and cryobiopsy in a hybrid operating room. Patient demographic characteristics, computed tomography images, rapid on-site evaluation cytology, and final pathology reports were collected. Forty-one patients underwent transbronchial biopsy and 6 received additional percutaneous transthoracic core-needle biopsy during the same procedure. The overall diagnostic yield was 88%. The complications included three patients with pneumothorax after receiving subsequent CT-guided percutaneous transthoracic needle biopsy, and two patients with hemothorax who underwent transbronchial cryobiopsy. Overall, the bronchoscopic biopsy of PPN using AFB and CBCT as precise guidance in the hybrid operating room is feasible and can be performed safely with a high diagnostic yield.

## 1. Introduction

Lung cancer is one of the most commonly diagnosed cancers worldwide, with over 2 million new cases and 1.8 million deaths in 2020. It is the leading cause of cancer-related mortality and morbidity [1]. Lung cancer has a poor prognosis; however, early detection can aid in better treatment and prognosis management [2]. Thus, low-dose computed tomography (CT) has been widely applied for lung cancer screening, and the detection of small indeterminate lesions has been rising. Therefore, this has further raised the need for efficient diagnostic tools and curative treatment [3].

Several methods have been adopted for the diagnostic sampling of lung lesions. Surgical resection (video-assisted thoracoscopic or open) is one of the most primitive diagnostic procedures; however, it is invasive and carries a high risk of complications [4]. Computed tomography (CT)-guided transthoracic needle aspiration is another method for navigating peripheral pulmonary nodules (PPN) with a higher diagnostic yield [5]. Nevertheless, transthoracic needle punctures have been associated with more complications, such as pneumothorax and hemorrhage [6,7]. Traditional bronchoscopy biopsy, guided only by fluoroscopy, presented fewer complications but a lower diagnostic yield and success rate [8].

Further advancements in navigation bronchoscopy techniques have allowed access to the PPN with higher accuracy and reduced complication rates. In addition, the procedure is conducted using a bronchoscope via a natural orifice, without traversing the pleura, thus making it less complicated and with a lower social risk [9]. The advent of endobronchial ultrasound (EBUS)-guided transbronchial needle aspiration has made it possible to perform a guided biopsy of small PPNs and lesions located near the airways [10]. However, the diagnostic yield of EBUS for extremely small PPNs is relatively unsatisfactory [11]. Tools such as a thin bronchoscope and guide sheath or techniques such as fluoroscopy or virtual bronchoscopy navigation have been developed to assist EBUS-guided biopsy in increasing diagnostic yield. Nevertheless, some limitations still demonstrate the necessity for more effective systems [11].

Cone-beam computed beam CT (CBCT) is a well-established imaging tool often used during a transbronchial biopsy for peripheral pulmonary lesions (PPL). CBCT has proven beneficial in navigating the target lesion, confirming the device tool-in-lesion, and in tissue acquisition for biopsy. In addition, it has been helpful in bronchoscopy ablation to track the location of the ablation probe relative to the target lesion [12]. One of the most recent developments in augmented fluoroscopic bronchoscopy (AFB) is using three-dimensional (3D) preoperative CT images overlaid with real-time fluoroscopy guidance for endobronchial navigation [13]. AFB presented with increased diagnostic yield and accuracy and a reduced procedural dose compared to previous fluoroscopic techniques [14,15]. A previous study that assessed the efficacy of EBUS with and without AFB for PPL biopsy showed an increase in navigation ability and a decrease in procedure time for patients who underwent EBUS with AFB compared to those who underwent EBUS only [16]. However, in these procedures, CBCT was not applied as a confirmation technique, considered the gold standard for checking the probe location. EBUS performed with AFB and CBCT is an advanced technique that is more efficient in reaching the target, with a higher diagnostic yield [11]. AFB with CBCT has advantages for interventional pulmonologists in complicated cases involving PPN. However, it may require a specialized setting and may not be readily available in many centers because of its high setup and maintenance costs [17,18].

In this study, we share the experience of the specialized setting of a hybrid operating room (HOR) for lung biopsy procedures. An HOR setting can provide real-time imaging and guide and confirm the navigation of difficult-to-access or small-sized lesions [17]. Furthermore, performing a procedure in an HOR helps reduce patient discomfort, avoids transportation of patients to different rooms, and reduces complications. However, few centers have adopted this approach in interventional pulmonology. Our center initiated these procedures in the HOR in 2020, which was built with the capacity to provide single-stage procedures and, most importantly, share the schedule and cost, thus adding to its economic utility. Even those with minimal experience can follow the system for successful lung marking because the equipment (CBCT-AFB) provides image guidance for bronchoscopy localization and percutaneous biopsy. Confirmation scanning and image reconstruction can also help thoracoscopic surgeons perform adequate resection [19].

This research aimed to share a single institute’s experience in assessing the feasibility and safety of bronchoscopy biopsy in an HOR via AFB and CBCT. In addition, we demonstrated some of these procedures that involve percutaneous biopsy in the HOR setting.

## 2. Materials and Methods

### 2.1. Patients

We recruited 41 consecutive patients who underwent diagnostic procedures for PPLs in the hybrid operating room of the National Taiwan University Hospital, Hsinchu branch, from July 2020 to August 2022. The patients underwent AFB-EBUS-guided transbronchial biopsy with or without additional CBCT-guided percutaneous transthoracic core needle biopsy. In addition, we acquired data from the database and medical records of the teaching hospital. The Institutional Review Board of the National Taiwan University Hospital, Hsin-Chu Branch (110-074-E) approved this study.

### 2.2. Procedures

All the procedures were performed in a hybrid operating room equipped with a robotic C-arm CBCT (ARTIS pheno, Siemens Healthcare GmbH, Forchheim, Germany). The patients underwent general anesthesia with single-lumen endotracheal tube intubation before the procedures. After that, each patient was placed in the supine position (Figure 1). CBCT images were obtained using a 4 s scan protocol during the end-inspiratory hold phase. CBCT imaging data were transferred to a nearby workstation (*syngo* X Workplace, Siemens Healthcare GmbH, Forchheim, Germany). The target pulmonary lesion was identified and marked using semi-automatic segmentation software (*syngo* iGuide Toolbox, Siemens Healthcare GmbH, Forchheim, Germany). The marked contours were then projected on a 2D fluoroscopy live screen following their corresponding 3D locations; thus, an augmented fluoroscopy system was configured. The bronchoscopy procedure was performed under the guidance of CBCT-augmented fluoroscopy, and a flexible bronchoscope (BF-1TQ290, BF-Q290, BF-P290 and BF-MP290F, Olympus, Tokyo, Japan; EB-530S and EB-530T, Fujifilm, Tokyo, Japan) was inserted through the endotracheal tube into the distal bronchus. After inserting the radial EBUS miniprobe through the working channel, its tip was adjusted under AF guidance. The C-arm fluoroscope was rotated to confirm the 3D position of the tip approaching the target lesions. For small PPLs (i.e., less than 2 cm in diameter), additional intraprocedural CBCT would be performed for more accurate localization. The bronchoscope was disconnected temporarily from the processor and placed on a specially designed holding bracket during the image acquisition of CBCT.

After confirming the location of the lesion using radial-EBUS imaging with or without intraprocedural CBCT, a transbronchial biopsy was performed with forceps (FB-231D.A; Olympus Co., Tokyo, Japan) for specimen collection. In addition, rapid on-site cytological evaluation (ROSE) was performed to confirm lesion access. The material from the forceps biopsy was imprinted on a clear glass slide and stained using a rapid method (Hemacolor; Merck KGaA, Darmstadt, Germany) for ROSE. If neither a malignant cell nor a specific finding was noted in the ROSE study, we would change to another site for biopsy. The procedure was terminated when there were no specific findings for more than three attempts or if the patient’s condition was not suitable for a further biopsy attempt. The histological samples were placed in 10% formalin, embedded in paraffin for routine histological evaluation with hematoxylin and eosin staining, and interpreted by cytopathologists. Tissue cultures of bacteria, mycobacteria, and fungi were also added when an infectious PPL was suspected. If the ROSE study showed inadequacy for diagnosis or the biopsy sample was insufficient for advanced studies, such as next-generation sequencing, we would perform additional transbronchial cryobiopsy (TBCB) to the targeted lesion if the risk of complications was acceptable. Another option was to shift the procedure to CBCT-guided percutaneous transthoracic core-needle biopsy. We obtained additional computed-tomography-like images using a C-arm system (DynaCT^®^, Siemens, Medical Solutions Erlangen, Germany), and the access path was laid out in the isotropic dataset using the *syngo* Needle Guidance of a *syngo* X Workplace (Siemens Healthcare GmbH). The needle path was defined by marking the entry and target points of the needle and subsequently projected with a laser beam on the patient’s skin. A needle biopsy was performed using a 17 G coaxial needle and an 18 G biopsy needle (Figure 2C,D).

### 2.3. Data Collection

We collected clinical data, including patient characteristics, features of the pulmonary lesions, and procedural details. Procedure time was defined as the time from bronchoscope insertion into the endotracheal tube to bronchoscope withdrawal from the endotracheal tube when the patient received only transbronchial biopsy. If an additional percutaneous biopsy was performed, the procedure ended with coaxial needle retraction from the skin. Total anesthesia time was defined as the time elapsed between the start of anesthetic induction and the extubation of the endotracheal tube. Global operation room time was defined as when the patient entered the hybrid operating room and left the hybrid operating room. The total accumulated radiation exposure dose expressed as the dose area product (DAP), was retrospectively calculated using data stored in the ARTIS workstation (Syngo Workplace).

We also assessed the complication rates, length of postoperative hospital stay, presumptive diagnosis, and final pathological diagnosis. Descriptive statistics were presented as medians with interquartile ranges (IQR) for continuous data and summarized as counts (percentages) for categorical data.

## 3. Results

The demographic characteristics of the patients and pulmonary lesions are summarized in Table 1 and Table 2. The findings from the preoperative CBCT categorized 22 (55%) nodules as pure ground-glass nodules, 4 (10%) as part-solid nodules, and 14 (35%) as solid nodules. The median nodule size (*n* = 31) was 35 mm, and the median nodule depth from the pleura was 12 mm. The remaining lesions (*n* = 10) were diagnosed as interstitial lung disease and were located close to the pleura.

Table 3 illustrates the details of AFB/CBCT-guided bronchoscopy biopsy. The median procedure time was 97 min, the median length of time under anesthesia was 146 min, and the median time in the operating room was 152 min. The median fluoroscopy duration for 25 patients with available radiation reports was 9.4 min. The median total dose area product for a single DynaCT scan was 6645 µGym^2^. Sixteen patients had missing radiation dose information. The median length of the postoperative stay was 2 days.

The details of the concurrent procedures, diagnoses, and complications are shown in Table 4. Additional EBUS-TBNA was performed in 10 patients (24.4%). Additional percutaneous biopsy was performed in six (14.6%) patients. The remaining 25 patients (61.97%) underwent transbronchial biopsy alone. The presumptive diagnosis revealed 17 (41.5%) benign lesions, and 24 (58.5%) malignant lesions. Among patients with malignant diagnosis, the diagnostic yield of the bronchoscopic biopsy was 87.8% (19 of 24). Complications such as pneumothorax were experienced by three (7.3%) patients who underwent an additional percutaneous biopsy, while hemothorax was observed in two (4.9%) patients who underwent additional percutaneous biopsy and transbronchial cryobiopsy, respectively.

## 4. Discussion

### 4.1. Comparison with Other Modalities

Our development of the AFB technique in HOR, which links the endobronchial road mapping of preprocedural CT and the target lesion derived from intraprocedural CBCT, presents a new platform for fluoroscopy-guided endobronchial intervention [20]. This new navigation technique provides the ease of visibility of small lung nodules and the location of lesions masked by other organs and differentiates AFB from standard fluoroscopic procedures [16]. Furthermore, this technique has also seen advantages in video-assisted thoracoscopic surgery for marking multiple pulmonary nodules without causing secondary pneumothorax [21]. Electromagnetic navigation bronchoscopy (ENB) is another popular bronchoscopic biopsy technique. It applies preoperative CT data and an electromagnetically tracked guide for procedural planning and guidance [22]. Although the complication risk is reduced, there are limitations to ENB, such as lower diagnostic yield.

Further advancements, such as robotic-assisted ENB, showed the feasibility and improved results in preliminary studies. However, research on a larger number of participants is needed [23]. ENB-guided biopsies using CBCT and AFB in an HOR also showed a diagnostic yield of 83.7% [22]. In our study, bronchoscopic biopsy using AFB and CBCT in HOR confirmed that endobronchial ultrasound had an overall diagnostic yield of 88% for both benign and malignant diseases.

Without the aid of fluoroscopy, a guide sheath was usually used for better diagnostic yield in transbronchial biopsy procedures [24]. However, small-sized forceps were required to pass through the sheath, which may have resulted in a smaller sample size. Real-time localization of PPLs with tissue sampling tools under augmented fluoroscopy further curtailed the need for a guide sheath and was seldom applied in this study. Thus, we could employ large-tissue forceps without the guide sheath, facilitating larger tissue sample acquisition. In addition, these image augmentation and projection techniques are well suited for the biopsy of small or sub-solid nodules or ground-glass nodules—those that are not visible under the standard fluoroscopy usually used to detect only large and solid lesions.

### 4.2. Settings of Robotic CBCT in Hybrid Operating Room

CBCT imaging is a relatively recent modality in interventional pulmonology. It is available as a floor-mounted, biplane, ceiling-mounted, or robotic C-arm system in interventional radiology or hybrid operating rooms [25]. Our previous study used a floor-mounted C-arm system [15] for lung marking and localization. However, the C-arm position in front of the patient’s head may not be optimal for the surgeon to conduct a bronchoscopy. Floor and biplane systems may restrict the area of rotation close to the patient’s head, thus limiting the reach for nodules located towards the base of the lung [12]. In this regard, ceiling-mounted systems are relatively easier to use than floor-mounted systems because they allow more space at the patient’s head and provide more flexible rotation during CBCT scanning (reducing the likelihood of collision) [25]. In this study, we used a robotic C-arm that can be placed at the head side (the most optimal position because of better rotational flexibility) or oblique side while tilting the C-arm to the left or right to reserve space for head movement. Robotic C-arms can be moved away from the patient without affecting the operator or nearby equipment [12]. It also provides more positioning flexibility and thus is optimal for diagnostic bronchoscopy.

### 4.3. Pros and Cons of General Anesthesia and Endotracheal Intubation during the Procedures

Furthermore, endotracheal intubation was performed in all patients, and the biopsy procedure was performed under general anesthesia (GA). This setup had several advantages. First, deep sedation helped control and achieve the desired respiratory movement and minimized the CT-to-body divergence. Endotracheal intubation may also be beneficial for securing the airway and facilitating better management of complications such as massive bleeding. For instance, a double-lumen endotracheal tube is separated into left and right bronchi. In cases where bleeding occurs on one side of the chest following a biopsy, a cuff can be placed around one end of the tube, and the balloon can be inflated to prevent blood flow to the other side of the chest. Tools such as bronchoscopes can pass through the secured airway to the lesion. In the case of percutaneous biopsy, one-lung ventilation may help limit lung movement during inspiratory breath holding, which can prevent pneumothorax. General anesthesia may also have some disadvantages. For instance, a long GA duration can lead to lung atelectasis (particularly in the dependent or posterior part of the lung), which may obscure the target lesion. However, this may also lead to a risk of hemodynamic instability, resulting in the use of inotropes and the need for prolonged intubation with ventilator support. It requires close monitoring as extubation cannot be performed immediately and may result in a prolonged recovery. In our study, the duration of GA was approximately 135 min and did not lead to any potential risks or complications from anesthesia. In addition, bronchoscopes need to pass through the limited space within the endotracheal tube, which may increase the risk of bronchoscopic surface damage because of repeated insertion and removal. In this study, this surface damage was prevented using a large endotracheal tube (8.0) and the application of lubricants on the outer surface of the bronchoscope.

### 4.4. Reasons for Additional Percutaneous Biopsies

In 14.6% of cases, additional percutaneous biopsies were performed for different reasons. Bronchoscopy biopsy may yield smaller specimens, which may require percutaneous methods, especially in cases of tissue fibrosis. However, this limitation can be overcome soon by newer techniques, such as cryobiopsy, which can acquire larger samples via bronchoscopy. EBUS is often used as a confirmatory protocol to ensure the location of lesions, for which CT bronchial signs aid in predicting the results. However, it is less effective in cases of non-visualization of a concentric lesion; thus, a percutaneous biopsy was considered. Following the bronchoscopy biopsy, a non-diagnostic rapid on-site cytological evaluation (ROSE) was performed in our hospital to confirm the tentative diagnosis made before the biopsy. If the evaluation result was negative or non-diagnostic (because of inadequate tissue imprint under the microscope), the sample was re-confirmed, for which we performed either an additional bronchoscopy or percutaneous biopsy to increase the diagnostic yield further.

### 4.5. Limitations

This study has some limitations. First, this study presented a single-institute experience with a limited number of patients. Second, there was no comparison with a control group; thus, details of the effect of the procedure were lacking. Finally, the current sample size was insufficient to establish proficiency in this technique. Future studies with larger numbers of patients are required to evaluate the protocol for this technique.

## 5. Conclusions

The advantages of HOR settings include robotic C-arm CBCT, which can easily acquire real-time intra-operative 3D images, an advanced working station that enables quick and accurate imaging, processing, and augmentation, and one-stop management of ROSE-confirmed malignant or probable malignant PPL ranging from percutaneous core needle biopsy to percutaneous microwave ablation or cryoablation. A multidisciplinary team discussion involving interventional pulmonologists, thoracic surgeons, radiologists, and cytopathologist is necessary for personalized planning before the intervention. In conclusion, our initial results of CBCT-AF-guided bronchoscopy biopsy are feasible for the diagnosis of not only lung cancer but also some challenging benign diseases, such as interstitial lung disease. In a HOR, this can be performed safely with a high diagnostic yield, acceptable radiation exposure and procedure duration, reduced length of hospital stay, and minimal complications.

## Figures and Tables

**Figure 1 diagnostics-13-01055-f001:**
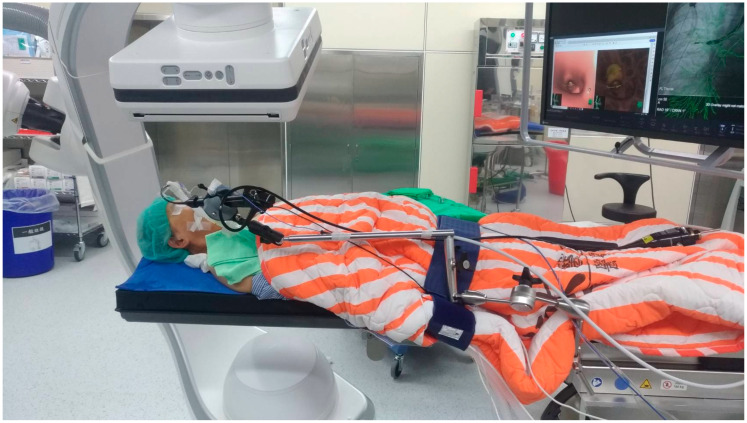
Bronchoscope setting on the designed bracket under robotic cone-beam computed tomography imaging.

**Figure 2 diagnostics-13-01055-f002:**
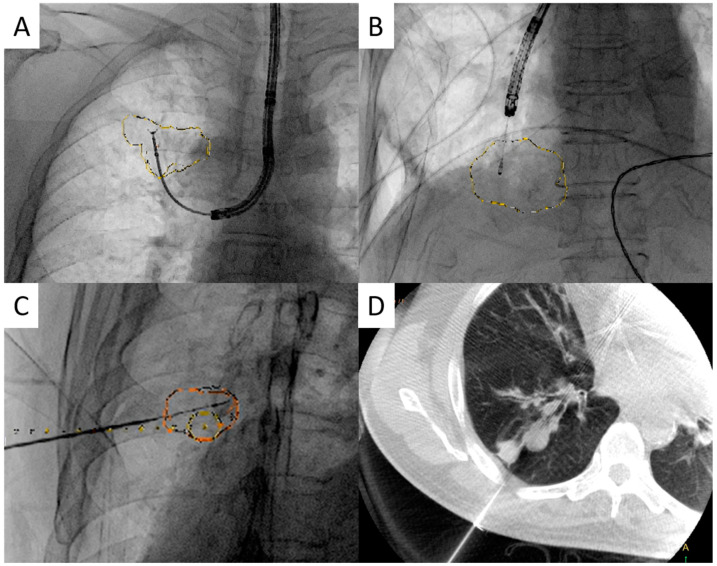
(**A**) Cone-beam computed tomography (CBCT)-derived augmented fluoroscopy (AF)-guided transbronchial forceps biopsy to the right upper lung peripheral pulmonary nodule (PPN); (**B**) CBCT-derived AF-guided transbronchial cryobiopsy to the right lower lung PPN; (**C**) CBCT-derived AF-guided percutaneous transthoracic core-needle biopsy with a coaxial needle to the right lower lung PPN; (**D**) CBCT confirmation of the relative position of the biopsy needle and the PPN.

**Table 1 diagnostics-13-01055-t001:** Characteristics of patients (*n* = 41).

Variables	Values (%)
Sex (female)	22 (53.7)
Age (y)	66 (58–76) *
BMI (kg/m^2^)	23.7 (20.4–26.2) *
ASA physical status classification	
Class I and II	20 (48.8)
Class III	20 (48.8)
Class IV	1 (2.4)
Smoking status	
Never smoker	27 (65.9)
Ex-smoker	6 (14.6)
Active smoker	8 (19.5)

* Continuous data are shown as median/interquartile range.

**Table 2 diagnostics-13-01055-t002:** Characteristics of pulmonary lesions (*n* = 41).

Variables	Values (%)
Lesion size (mm) (*n* = 31 ^a^)	35.0 (17.0–44.0) *
<10 mm	1 (3.2)
10–20 mm	8 (25.8)
20–30 mm	1 (3.2)
30–40 mm	8 (25.8)
40–50 mm	11 (35.5)
>50 mm	2 (6.5)
Lesion distance from the pleura (mm) (*n* = 31 ^a^)	12.0 (0.0–22.0) *
<20 mm	22 (71.0)
20–40 mm	7 (22.6)
>40 mm	2 (6.4)
Nodule appearance	
Pure GGN	5 (12.2)
Part-solid GGN	14 (34.1)
Solid	22 (53.7)
Bronchial sign	
Positive	33 (80.5)
Negative	8 (19.5)
Location	
Right upper lobe	14 (35.0)
Right middle lobe	2 (5.0)
Right lower lobe	9 (22.5)
Left upper lobe	9 (22.5)
Left lower lobe	7 (17.5)

^a^ Patients had interstitial lung disease with diffuse lesions, which made the measurement of the lesion size and distance from the pleura not applicable. ASA, American Society of Anesthesiologists; CT, computed tomography; GGN, ground-glass nodules. * Continuous data are shown as median/interquartile range.

**Table 3 diagnostics-13-01055-t003:** Details of AFB/CBCT guided bronchoscopy biopsy (*n* = 41).

Variables	Values ^a^
Procedure time (min)	97.0 (64.0–122.0)
Length of time under anesthesia (min)	146.0 (124.0–193)
Global operation room time (min)	152.0 (130.0–197.0)
Radiation reports (*n* = 25)	
Number of DynaCT scans	2 (2–3)
Duration of fluoroscopy (min)	9.4 (5.1–11.7)
Total dose area product (µGym^2^)	6645 (5453–8326)
Length of postoperative stay (day)	2.0 (1.0–4.0)

^a^ Present as median (interquartile range).

**Table 4 diagnostics-13-01055-t004:** Diagnostic yields, complications, and additional percutaneous biopsy.

	Number (%)
Transbronchial biopsy	25 (61.97)
Concurrent procedures	
Percutaneous biopsy	6 (14.6)
Transbronchial needle aspiration	10 (24.4)
Diagnosis	
Presumptive benign diagnosis	17 (41.5)
Acute and chronic inflammation	1 (2.4)
Chronic inflammation	5 (12.2)
Chronic inflammation and interstitial fibrosis	7 (17.1)
Necrosis	1 (2.4)
Peribronchiolar metaplasia	1 (2.4)
Benign peribronchial tissue	2 (4.9)
Presumptive malignant diagnosis	24 (58.5)
Non-small-cell carcinoma	14 (34.1)
Small cell carcinoma/neuroendocrine tumor	2 (4.9)
Metastasis	1 (2.4)
Atypia	2 (4.9)
Negative for malignancy	5 (12.2)
Complications	
Pneumothorax ^a^	3 (7.3)
Conservative management	1 (2.4)
Simple drainage	2 (4.9)
Bleeding (hemothorax) ^b^	2 (4.9)

^a^ All three patients underwent percutaneous biopsy in the same procedure. ^b^ One patient underwent percutaneous biopsy using the same procedure, and the other underwent transbronchial cryobiopsy.

## Data Availability

All data generated or analyzed during this study are included in this published article.

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
