# Peer review of "Cone-Beam Computed-Tomography-Derived Augmented Fluoroscopy-Guided Biopsy for Peripheral Pulmonary Nodules in a Hybrid Operating Room: A Case Series"

_diagnostics, 2023, doi:10.3390/diagnostics13061055_

Round 1
Reviewer 1 Report
I have the following comments:
-Abstract. Please report the number of patients enrolled (line 21). Moreover, the sentence at lines 26-27 ('The overall diagnostic procedure yield and patient rate were 88% and -related to pneumothorax and bleeding, respectively') is unclear and should be rephrased.
-Introduction. Please remove the underlining at lines 92-93.
-Materials and methods.
a) Line 97. Again, please report the number of consecutive patients enrolled.
b) Line 100. Please explain the meaning of the "iGuide" percutaneous needle biopsy. I suppose this is a commercial name - in case, the procedure should be described in detail and the commercial name put in parentheses along with the vendor's name.
c) Please briefly report the CBCT image acquisition parameters (kV, mAs, etc.), and explain (line 140) the meaning of the term 'DynaCT'.
Author Response
Thanks for your review.
Here's our reply:
- The abstract had been revised.
- The underlining at lines 92-93 had been removed.
- a) 41 patient; b) The "iGuide" had been replaced by CBCT-guided percutaneous transthoracic core needle biopsy; c) The term 'DynaCT' had been explained in detail.
Reviewer 2 Report
Manuscript Title “Cone-Beam Computed Tomography Derived Augmented Fluoroscopy-Guided Biopsy for Peripheral Pulmonary Nodules in a Hybrid Operating Room: A Case Series”
General comment:
The quality of this manuscript is OK in either structure or text flow, however, the shown data in the result are so domestic that cannot be verified elsewhere and the description in the discussion need to be further elaborated to express the findings according to this auxiliary technique. Some specific comments are listed below in various section
Specific comment:
1. Abstract; too few the technical term to draw a general idea of this technique. The background of general review in the first section can be eliminated and increased the quantified results instead
2. Introduction; L80, the specific description of the technical process can be elaborated more.
3. Materials and methods; 2.1 an additional table of patients’ characteristics is suggested to add here for providing more specific information
4. Procedure; too messy to describe the process, suggest to add a flowchart for implying the solid process step by step
5. Figure 1; too messy to lose the focus, can it be replaced with neater one without the complicate background?? Is it an empty container nearby?? What for??
6. Table2; the format is too messy to imply the derived result, suggest to revise thoroughly with additional unit and quantified data with statistical definition
7. Discussion; add the subsection to clearly imply the focus of the following statements
8. Suggest to summarize the discussion into an easy understanding table and elaborate the description around the listed data, otherwise, it is easy to lose the focus of point.
9. Conclusion; too short to imply a strong conclusion, 150 words maybe a good suggestion to conclude the finding
Author Response
Thanks for your great suggestion.
The patient number was not large and the patient's detailed characteristics were so varified which may make the characteristics of PPL out of focus.
The table 2 had been simplified.

Round 2
Reviewer 2 Report
Manuscript Title “Cone-Beam Computed Tomography Derived Augmented Fluoroscopy-Guided Biopsy for Peripheral Pulmonary Nodules in a Hybrid Operating Room: A Case Series”
General comment:
The quality of the revised manuscript is not OK. The author does not revise the content follow the suggested comments, especially the discussion, there is no revision at all.
Specific comment:
1. Abstract; still too few the technical description in the abstract, and some redundant background review still exists to prolong the length
2. Materials and methods; still no table to summarize the patients’ characteristics
3. Figure 1; still suggest to replace a new figure, the original one is too messy
4. Discussion; still no subsection to clearly imply the focus of the following statements
5. Strongly suggest to summarize the discussion into an easy understanding table and elaborate the description around the listed data, otherwise, it is easy to lose the focus of point.
6. Conclusion; still too short to imply a strong conclusion, 150 words maybe a good suggestion to conclude the finding
Author Response
- Abstract; still too few the technical description in the abstract, and some redundant background review still exists to prolong the length
Response: Thanks for your great recommendation. The abstract content had been enriched.
- Materials and methods; still no table to summarize the patients’ characteristics
Response: Thanks for your great recommendation. We had added Table 1 to summarize the patients’ characteristics.
- Figure 1; still suggest to replace a new figure, the original one is too messy
Response: Thanks for your great recommendation. We had replaced the figure with a new one.
- Discussion; still no subsection to clearly imply the focus of the following statements
- Strongly suggest to summarize the discussion into an easy understanding table and elaborate the description around the listed data, otherwise, it is easy to lose the focus of point.
Response: Thanks for your great recommendation. We had divided our discussion part into subsection and also done some modification of the content.
- Conclusion; still too short to imply a strong conclusion, 150 words maybe a good suggestion to conclude the finding
Response: Thanks for your great recommendation. We had increased the content of conclusion.
Round 3
Reviewer 2 Report
the revised version is OK to accept